# Cognitive Reserve and Its Association with Cognitive and Mental Health Status following an Acute Spinal Cord Injury

**DOI:** 10.3390/jcm12134258

**Published:** 2023-06-25

**Authors:** Mohit Arora, Ilaria Pozzato, Candice McBain, Yvonne Tran, Danielle Sandalic, Daniel Myles, James Walter Middleton, Ashley Craig

**Affiliations:** 1The Kolling Institute, Faculty of Medicine and Health, The University of Sydney, Sydney, NSW 2000, Australiadanielle.sandalic@health.nsw.gov.au (D.S.); a.craig@sydney.edu.au (A.C.); 2John Walsh Centre for Rehabilitation Research, Northern Sydney Local Health District, St Leonards, NSW 2065, Australia; 3Australian Institute of Health Innovation, Macquarie University, North Ryde, NSW 2113, Australia

**Keywords:** spinal cord injury, cognitive impairment, cognitive reserve, depression, fatigue, rehabilitation

## Abstract

Background: Mild cognitive impairment (MCI) is a common secondary condition associated with spinal cord injury (SCI). Cognitive reserve (CR) is believed to protect against cognitive decline and can be assessed by premorbid intelligence (pmIQ). Despite the potential utility of pmIQ as a complementary metric in the evaluation of MCI in SCI, this approach has been infrequently employed. The purpose of this study was to examine the association between MCI and pmIQ in adults with SCI with the aim of exploring the potential value of pmIQ as a marker of CR in this population. Methods: Cognitive function was assessed on three occasions in adults with SCI over a 12-month period post-injury, and pmIQ was assessed once at baseline. Demographic and mental health measures were also collected, and logistic regression was conducted to determine the strength of association between pmIQ and MCI while adjusting for factors such as mental health and age. Results: The regression analysis revealed that at the time of admission to SCI rehabilitation, the MCI assessed by a valid neurocognitive screen was strongly associated with pmIQ. That is, if a person has MCI, there was 5.4 greater odds (*p* < 0.01) that they will have poor pmIQ compared to a person without MCI after adjustment for age and mental health. Conclusions: The assessment of CR is an important area that should be considered to improve the process of diagnosing MCI in adults with an acute SCI and potentially facilitate earlier intervention to slow or prevent cognitive decline.

## 1. Introduction

Spinal cord injury (SCI) is a severe neurological injury resulting in substantial impairment, including loss of motor control and sensory and autonomic dysfunction [1]. Common physical secondary conditions include cardiovascular and respiratory dysfunction, pain, pressure injuries, urinary tract infections, and bowel complications, while psychological-based problems include fatigue, cognitive impairment, and mental health disorders [1,2,3,4,5,6]. The occurrence of mild cognitive impairment (MCI) after SCI is considered to be high, with best estimates suggesting rates of between 20 and 30% in the acute SCI adult population [4,7]. However, recent studies have highlighted a number of issues when assessing MCI after SCI [4,7,8]. These issues include: (i) broad estimates of the occurrence of MCI (ranging between 10–60%), derived from cross-sectional research conducted over the past 20 years; (ii) the lack of clarity about which cognitive domains are most affected by SCI [4]; (iii) the absence of a universally accepted definition for the identification of cognitive impairment following SCI [7]; (iv) the lack of a neurocognitive screen that has demonstrated structural validity for assessing MCI after SCI [7]; (v) the lack of cognitive performance norms for assessing MCI after SCI [7]; (vi) the lack of clarity around the contribution of environmental factors to cognitive performance during the acute and rehabilitation phases, and (vii) the problem of misdiagnosing cognitive impairment resulting from the application of norm-referenced assessments with little regard for measures of baseline or premorbid cognitive intelligence/cognitive reserve [8]. While these problems have been deliberated upon [4,7,8], they have not been resolved and therefore pose significant barriers to SCI adjustment and rehabilitation outcomes. This paper will specifically investigate the concerns posed by the seventh issue; that is, the importance of cognitive reserve (CR) following SCI and its relationship with post-cognitive performance, demographics such as age and education, and psychosocial factors, including depressive mood, anxiety, and fatigue.

The theory of CR refers to the brain’s ability to maintain cognitive function despite age-related changes or brain damage and addresses the observation that people differ in how they fare after neurological injury [9,10]. For example, people with higher levels of CR may be less likely to experience cognitive decline or develop dementia as they age [10]. CR is based on the assumption that individual differences in post-neurological injury recovery are associated with cognitive abilities developed over the life span, which are believed to protect against and compensate for neurological damage caused by a traumatic injury (e.g., traumatic brain injury/TBI) or a disease process (e.g., Alzheimer’s disease) [9,10]. Evidence suggests CR has strong convergent validity and reasonable discriminant validity and, therefore, can be considered a valid cognitive concept [9]. Evidence also suggests factors such as education and engagement in social and occupational activities increase CR, and these factors could therefore be used as measures of CR [9,10]. Premorbid intelligence is another important estimate of CR. An acceptable measure of premorbid intelligence (pmIQ) is the Test of Premorbid Functioning (TOPF), an objective lexical task. The premise is that vocabulary is highly correlated with cognitive function and, therefore, is considered a valid measure of global intelligence [8,9,10,11,12]. For example, research in individuals with a TBI found higher levels of premorbid intelligence (assessed by TOPF), and thus an assumed higher CR was related to more favourable neuropsychological outcomes [13]. We contend that diagnosing MCI based solely on population norms without recourse to the assessment of premorbid intelligence limits its application in SCI rehabilitation [8]. As an illustration of this assertion, in a study involving simulated data for 500,000 adults with SCI, norm-referenced and premorbid-intelligence methods of MCI screening were compared to examine the extent of MCI misclassification after SCI [8]. Results found that up to 20% of the simulated adults with SCI were potentially misclassified as having MCI. The study concluded that measures of premorbid intelligence (e.g., TOPF) should be included in the assessment of cognitive function after SCI in the absence of baseline and longitudinal cognitive screen measures [8].

Success of SCI rehabilitation is dependent to a certain extent upon the client having sufficient cognitive capacity to understand the complex instructions concerning medications, medical procedures, self-management, and ongoing self-care [1,7,8]. It is also crucial that these self-management skills learned in the rehabilitation phase can be translated into the community setting, such as managing caregivers when required. For this to occur, an evidence-based and effective system for classifying MCI is required. Therefore, the main aim of this study was to investigate the relationship between CR/pmIQ (assessed by TOPF) and post-SCI cognitive outcomes (assessed by the Neuropsychiatry Unit Cognitive Assessment Tool, NUCOG), as well as examine relationships between pmIQ and additional factors including age, education, level of injury, mental health, and fatigue. The main hypothesis is that TOPF scores will be strongly associated with neurocognitive scores post-SCI. Specifically, it was hypothesised that: (i) higher age, higher level of injury, lower years of education, lower pmIQ, higher depressive mood and anxiety, and higher fatigue will be related to lower neurocognitive scores; (ii) age, level of injury, and years of education will be related to CR (i.e., TOPF standard scores); (iii) adults with acute SCI who have lower neurocognitive scores after SCI and are therefore more likely to have MCI, will more likely have low scores on TOPF (i.e., low CR or pmIQ); (iv) psychosocial factors will influence the association between neurocognitive scores and TOPF scores assessed at admission to rehabilitation; so that those who have higher levels of depressive mood, anxiety, and fatigue will more likely have lower scores on TOPF (i.e., low CR or pmIQ).

## 2. Materials and Methods

### 2.1. Design

Full details of this study’s protocol have been published [14]. The design involved an inception cohort longitudinal study that followed adults with an acute SCI from the first 24–48 h after their presentation to a hospital emergency department to discharge from rehabilitation (usually 4–6 months post-SCI) and up to 12 months post-SCI when living in the community.

### 2.2. Participants

Participants with SCI (*n* = 75) were recruited when engaged in SCI rehabilitation in one of the three specialised SCI units in Sydney, New South Wales, Australia. Inclusion criteria consisted of: (i) aged between 17 and 80 years; (ii) having an acute SCI of non-traumatic or traumatic origin; and (iii) having proficiency in English enabling them to complete all assessments. Exclusion criteria consisted of: (i) the existence of a severe mental disorder (e.g., bipolar disorder or psychosis); and (ii) the existence of a severe pre-morbid (i.e., presence of a serious medical condition that existed before the onset of an SCI) or concurrent severe TBI (i.e., loss of consciousness > 24 h, post-traumatic amnesia > 7 days, or a Glasgow Coma score of 3–8 usually assessed within 24 h of initial injury). Full compliance with the Code of Ethics of the World Medical Association occurred, and the local institutional human research ethics committee granted ethics approval. All participants provided informed consent prior to participating. 

### 2.3. Study Measures

The protocol for this study provides full details of all the measures employed [14]. Participants were assessed three times (i.e., admission, discharge, and 12 months following SCI) during the study using multiple assessments, including cognitive and psychosocial measures. Baseline and discharge assessments were performed in one of three SCI units, and assessments at 12 months were performed in the community. However, only a limited number of measures believed to be relevant to the specific aims of this study are reported here. Socio-demographic measures included age, sex, and years of education. Injury characteristics included the level of injury and the American Spinal Injury Association Impairment Scale assessed by a medical specialist based on the International Standards for Neurological Classification of SCI (http://ais.emsci.org/, accessed 10 April 2023).

#### 2.3.1. Neuropsychiatry Unit Cognitive Assessment Tool

Cognitive function was assessed by the NUCOG [7,15,16]. The NUCOG is a validated neurocognitive screen comprising 21 items in 5 cognitive domains: attention, visuoconstructional, memory, executive, and language. Scores for each of the domains range between 0 and 20, adding up to a total NUCOG score out of 100. Higher scores indicate higher levels of cognitive function.

Neuropsychologists and psychiatrists developed the NUCOG based on multiple neurocognitive tests, including the Stroop, Trail Making Test, and WAIS-4th Edition. The NUCOG has been shown to have criterion, convergent, and discriminant validity for a number of disorders, and to have acceptable reliability and specificity/sensitivity [16]. When applied to screening adults with SCI who have restricted hand function, it was necessary to adapt some items (e.g., drawing a reproduction) as per previous studies, where this has been shown to not alter the validity of NUCOG scores [7,15]. When face-to-face administration of the NUCOG was not possible (e.g., social distancing restrictions due to COVID-19), administration occurred via telehealth. Administration of the NUCOG via teleconferencing required additional adjustments to the NUCOG assessment procedures to satisfy the telehealth environment [7,15]. These procedural adjustments have been discussed elsewhere [7,15]. Although the NUCOG has been shown to have poor structural fit in the Memory and Language domains in an SCI sample, the overall NUCOG score has demonstrated adequate validity [7].

While the definition and criteria for determining the presence of MCI vary [7], in this study, to determine the possible presence of MCI, the total NUCOG mean (reported elsewhere; [15,16]) was used, with probable MCI defined as one standard deviation (SD) below a mean of 92.9 (SD = 4.9). Therefore, the MCI cut-off score of 88 (93 mean score minus 5 as one SD) was used, so that those scoring ≤ 88 were considered to have a probable MCI, while those scoring > 88 were considered not to be cognitively impaired. 

#### 2.3.2. Test of Premorbid Functioning

The TOPF was assessed only at admission and was employed to estimate pmIQ/CR [8,11,13]. Completing the TOPF involved reading a list of 70 phonemically irregular words and was scored according to its manual instructions. The TOPF score consisted of the number of words the participant read aloud with the correct pronunciation to derive the raw score, which was then transformed into age-corrected standard scores provided by the test [11]. The TOPF has demonstrated validity [9,10]. The TOPF has a range of scores from 40 to 160. A score of 101 is considered average, while scores above 101 indicate above-average cognitive functioning (superior CR), and scores below 101 indicate below-average cognitive functioning [17].

#### 2.3.3. Patient Health Questionnaire—9 Items

Mood was assessed by the Patient Health Questionnaire-9 (PHQ-9) [18]. The PHQ-9 assesses each of the nine Diagnostic and Statistical Manual of Mental Disorders Fourth Edition (DSM-IV) criteria for major depressive disorder over the prior two-week period. Items range from 0 “not at all” to 3 “nearly every day” and are summed to establish depressive-symptom severity (0–4 no depressive symptoms; 5–9 mild depressive symptoms; 10–14 moderate depressive symptoms; 15–19 moderately severe depressive symptoms; and 20–27 severe depressive symptoms) [18]. The PHQ-9 has demonstrated validity [18,19].

#### 2.3.4. Generalized Anxiety Disorder—7 Items

Anxiety was assessed using the Generalized Anxiety Disorder-7 (GAD-7) [20]. The GAD-7 is a seven-item psychometric instrument used to determine the severity of generalized anxiety disorder (GAD). Participants are asked to rate their anxiety symptoms as experienced over the prior two weeks. Responses include “not at all (0)”, “several days (1)”, “more than half the days (2)”, and “nearly every day (3)”. The scores of the seven items are summed to provide a total score. Total scores of 5, 10, and 15, respectively, represent cut-points for mild, moderate, and severe anxiety symptoms [20]. The GADS-7 has demonstrated validity [19,20].

#### 2.3.5. Fatigue Severity Scale

Fatigue was assessed by the Fatigue Severity Scale (FSS) [21]. The FSS is a 9-item self-report scale that assesses the severity of fatigue and its effects on daily activities. Items are scored on a 7-point Likert scale (1 = strongly disagree to 7 = strongly agree). The higher the score, the more severe the fatigue. The FSS has demonstrated reliability and validity [21,22].

### 2.4. Analyses

Summary statistics were produced for the study variables. Pearson correlation analyses were conducted to determine relationships between age, sex, years of education, level of injury, and NUCOG scores with TOPF. Missing data for the TOPF and NUCOG variables were managed by listwise deletion of those participants who failed or could not be contacted to complete assessments in the second (discharge) and third (12 months) NUCOG assessments. Ten participants did not provide all NUCOG assessments or a TOPF assessment, leaving sixty-five participants for the logistic regression. For a small number of cases (less than 10%), item mean imputation was used for missing values in the psychosocial variables (GADS-7, PHQ-9, and FSS). The value of this technique is that it preserves statistical power but has less negative influence on variation than mean imputation (e.g., reducing standard error rates) than overall mean imputation [23]. Statistical power to find valid associations using multiple regression was calculated to be 98% given *n* = 65, α = 0.05, number of predictors = 5, and an estimated effect size of 0.25 [24]. Crude odds ratio, sensitivity, and specificity values were calculated from the contingency table of MCI versus no MCI as a function of low TOPF (≤101) versus high TOPF (>101).

Logistic regression with the dependent variable being the TOPF standardised score, dichotomised to include those scoring lower on the TOPF standardised score (≤101) and those scoring higher (>101). A standardised TOPF score of 101 was selected to include those who scored just over 100 in the low TOPF sub-group, as arguably, a score of 100 or just over should not be considered superior CR [17].

Five independent factors were entered into the logistic regression. Given the sample size available for the regression analysis (*n* = 65), the number of independent variables able to be entered was limited to five [23]. Total NUCOG scores assessed at admission were entered in dichotomous form, that is, probable cognitive impairment (≤88) versus probable no cognitive impairment (>88). Four continuous predictors were entered, consisting of age, PHQ-9, GADS-7, and FSS. The independent factors entered into the logistic regression were chosen because they have a theoretical basis for contributing to premorbid intelligence. For example, increased age (within the 17–80 age range) will be related to CR/ TOPF scores. Further, as higher pmIQ or CR is thought to be associated with better psychological outcomes, mental health measures were also included. Sex, years of education, and level of injury were not entered into the regression given restrictions on the number of independent variables. Restricting the number of independent factors entered into the regression complied with rules governing participants versus the number of independent variables entered [23]. Due to the limited sample size, logistic regression analysis was only conducted for admission to SCI rehabilitation. All analyses were performed using Statistica Version 13 (https://www.statistica.com, accessed 10 February 2023).

## 3. Results

Table 1 shows summary statistics and breakdown for the participants for socio-demographic, injury, TOPF, NUCOG, mental health, and fatigue factors. Table 1 shows participants classified with MCI had significantly fewer years of education (*p* < 0.05) and lower TOPF scores (*p* < 0.01). Participants classified as having MCI had significantly higher levels of depressive mood (*p* < 0.05) and anxiety (*p* < 0.01), though there were no differences in fatigue levels between these two sub-groups. For paraplegia, 17.6% (6 out of 34) had MCI, while 30.8% (12 out of 39) of those with tetraplegia had probable MCI (two missing values); however, this difference was not significant (Χ^2^ = 1.7, df = 1, *p* > 0.05). No differences were found for age or sex between those with and without MCI. 

### 3.1. Probable Rate of MCI

Based on *n* = 75 and the criteria employed for probable MCI in this study (NUCOG scores ≤ 88), the rate of MCI in this sample was 25.3% (19 out of 75) at admission to rehabilitation and 22% (9 out of 41) at discharge from rehabilitation (around 4–6 months post-SCI). Rates at 12 months are not reported given the low sample size at the 12-month assessment (ongoing data collection). 

### 3.2. Probable Diagnostic Classification of MCI

Table 2 shows a contingency breakdown of the classification of probable MCI as well as misclassifications based on the MCI and TOPF criteria. Odds ratios that are greater than 1 indicate that the event is more likely to occur as the predictor increases. The crude odds ratio (i.e., not adjusted for potential factors that could influence the ratio) was calculated to be 4.85, suggesting that if a person with acute SCI has a NUCOG score > 88 (not cognitively impaired), then there is 4.85 greater odds that they will have a TOPF score of >101 (superior CR), compared to the odds of such a relationship in a person with a NUCOG score of ≤88 (cognitively impaired). The reverse relationship holds. This finding indicates there is a strong positive association between post-SCI cognitive outcomes and CR (or premorbid intelligence as assessed by TOPF). Sensitivity refers to the ability of the TOPF score, say ≤101, to also have a NUCOG score ≤ 88 or vice versa. The sensitivity, as shown in Table 2, is moderate at 71.4%, indicating a number of false negative results exist (*n* = 17 or 63% misclassification). Specificity refers to the ability of the TOPF score to be more likely associated with those who do not have MCI. The specificity, as shown in Table 2, is also not high at 66%; however, there are fewer false positives (*n* = 4 or 10.8%). 

### 3.3. Association of Cognitive Reserve with Multiple Independent Factors in SCI 

Table 3 shows Pearson correlations between the TOPF standard score (estimate of CR) and socio-demographic, injury, NUCOG, and predictor factors. 

Table 4 shows the results of the logistic regression between TOPF and NUCOG, age, PHQ-9, GADS-7, and FSS. The objective was to determine the adjusted association between CR (TOPF: ≤101 versus >101) and the probable presence of MCI (i.e., ≤88 versus no MCI > 88), age, mental health, and fatigue at admission. Table 4 shows, after including these variables, that a strong positive association exists between NUCOG and TOPF, with an odds ratio of 5.4 (*p* = 0.05). Odds ratios less than 1 indicate that it is more likely that as the independent factor increases, TOPF will be reduced (i.e., ≤101). Therefore, as age increases, the odds increase that CR decreases (odds ratio = 0.95, *p* = 0.004). That is, a 5% decrease in the odds of a TOPF > 101 with each additional year (0.95–1 × 100 = −0.05 × 100 = 5%). For depressive mood, the same association was found with an odds ratio of 0.76 (close to significance at *p* = 0.06), suggesting that as depressive mood increases, the odds are that TOPF scores will be lower. In contrast with the findings of correlation analyses, anxiety (GADS-7) was found to have a positive odds ratio of 1.69 with TOPF, suggesting that if a person with acute SCI has higher anxiety, there is 1.65 greater odds that they will have a TOPF > 101, compared to the odds of such a relationship in a person with a TOPF score ≤ 101. The relationship between TOPF and fatigue was also less than 1 (odds ratio = 0.98, *p* = 0.57) but was not significant. 

## 4. Discussion

This study aimed to investigate the relationship between CR, as measured using the TOPF, and post-SCI cognitive outcomes using the NUCOG. The study also examined the potential value of CR as a marker of cognitive function in this population and explored relationships between CR and other factors such as age, education, level of injury, mental health, and fatigue. Our findings have important implications for improving the diagnosis and management of MCI in adults with SCI and suggest that assessing CR should become a standard component of evaluating neurocognitive status in this population.

Our hypothesis 1 was partially confirmed. Those with low neurocognitive scores (suggesting the presence of MCI) had an increased psychosocial burden [15]. Specifically, participants with MCI had significantly higher levels of depressive mood and anxiety but not fatigue. Participants with probable MCI also had fewer years of education and lower levels of CR compared to those without MCI. This finding confirms prior research that shows the presence of MCI is associated with an increased risk of secondary conditions such as depressive mood [15] and cardiovascular dysregulation [25]. Fatigue is a known problem in SCI [6], and our finding suggests it remains a challenge regardless of the presence of MCI. Past research has rarely found that demographic variables such as sex and injury variables are significantly related to neurocognitive scores [15]. Age has been found to be mildly related to MCI in a community sample of adults with SCI [15], but age was not significantly different between the MCI versus non-MCI sub-groups in this acute sample.

Prior research has shown that adults with SCI have problems in neural inhibitory function and perceptual encoding processes, as well as in executive functioning important for engaging in daily tasks [26]. As argued previously, adjusting to SCI, and understanding rehabilitation processes requires intensive learning and the use of adaptability skills to manage complications and improve adjustment to SCI. Deficits in cognitive performance [15,26] can impede outcomes, such as the use of computer assistive technology in the home environment [27], which can improve quality of life if used appropriately [28]. Therefore, it is important that neurocognitive screening be performed regularly after acute SCI, especially during intensive rehabilitation, to improve the detection rates of those with MCI. Strategies can then be developed to counter the negative influences associated with MCI [15].

Rates of MCI in adults with SCI have been shown to vary widely from 20–60% [4,7]. This variation is due to methodological issues, such as a lack of a universally accepted MCI definition, as well as assessments being conducted at different times post-SCI in past studies [4,8]. In the present study, a conservative definition of MCI was purposely employed; that is, MCI was defined as one SD below the mean for the population, and the rate was assessed at admission to rehabilitation just weeks after the injury. The MCI rate was found to be 25.3%. A slight decline in the MCI rate (22%) was found when MCI was assessed 4–6 months later when participants were discharged from rehabilitation into the community. There is not yet sufficient 12-month data to provide a reliable MCI rate. Nevertheless, this longitudinal data will be vital information for advancing how MCI is understood and managed during SCI rehabilitation. For example, a one-off neurocognitive assessment does not provide information about whether the person assessed is declining or improving in their cognitive capacity [8]. To the best of our knowledge, there have been no longitudinal/prospective reports of neurocognitive function in adults with SCI. One suggestion to enhance the diagnostic accuracy of MCI is to incorporate multiple assessment methods. For example, combining cognitive tests with other techniques such as neuroimaging (e.g., MRI) or biomarkers (e.g., blood-based markers) can provide a more comprehensive evaluation of cognitive functioning post-SCI. This approach aims to improve the overall accuracy of MCI diagnosis by considering various aspects of cognitive impairment.

The crude odds ratio from Table 2 confirms our hypothesis that CR (i.e., TOPF scores) is strongly associated with post-SCI cognitive outcomes. As previously discussed, relying on normative values to classify MCI has its limitations, and thus the significance of this finding is contingent upon the establishment and evaluation of a universally accepted definition for MCI that can be used reliably. Nevertheless, the crude odds ratio of 4.85 reported in Section 3.2 suggests a strong relationship exists between neurocognitive scores and CR. One can conclude from this that if a participant has a low NUCOG score suggesting MCI, then the odds are high (4.85%) that they will also have low CR (i.e., a low TOPF score) compared to a participant who has a high NUCOG score. This is similar to CR research in areas such as TBI and dementia, which has found people with low CR have an increased occurrence of having poorer neurocognitive outcomes [9,10].

However, as highlighted in prior research [8], there remains a risk of misclassification of diagnoses [8,12]. In this study, 32.8% of cases were potentially misclassified, with 62.9% being false positives (17 out of 27) and 10.8% (4 out of 37) being false negatives. The sensitivity of a test refers to its capacity to identify an individual as positive when they are, with suggestions that an acceptable sensitivity should be at least 80% [29]. A very sensitive test suggests there should be few false negatives and, therefore, fewer true cases are missed [29]. In this study, the sensitivity was not high at 71.4%, with almost a third of cases being false positives. Various factors could have contributed to the high false positive rate, such as those who do not have English as their first language, and thus, the TOPF score could be biased toward a lower score. Further research is required to explore other possible contributors to false positives. The specificity of a test is concerned with how capable it is of correctly identifying people who are truly negative [29]. While the test had a low number of false negatives, it has also been suggested that an acceptable specificity is around 80% [29]. Contributors to false negatives likewise need further investigation. Strategies are needed to resolve the misclassification of MCI post-SCI, such as improving sensitivity and specificity. For example, employing prospective designs involving multiple assessments of neurocognitive data over time post-SCI and simultaneously assessing relationships to alternative measures such as pmIQ alongside norm-based neurocognitive assessment at different stages post-SCI [8]. 

The Pearson correlation analyses revealed that CR (TOPF standardised score) was positively and significantly associated with age. It should be remembered that the age range investigated in this study was between 17 and 80 years, with a mean age of around 51 years. Therefore, there will be less influence from decreased CR due to older age. Perhaps within this age range, older participants have accumulated greater CR, or the positive crude correlation between age and TOPF (r = 0.52) does not take into account the possible influence of other factors on this relationship between age and TOPF. Research is required to clarify the positive correlation between age and TOPF. TOPF was also positively associated with higher years of education, confirming prior research that found education to be a CR protective factor [10,12]. TOPF was also positively and significantly associated with NUCOG scores at admission, discharge, and after 12 months post-SCI; again, in line with prior research in which higher CR is positively related to better neurocognitive outcomes post-injury [10,13]. As hypothesised, TOPF was negatively correlated with depressive mood (PHQ-9) and anxiety (GADS-7), though the relationship with PHQ-9 was not significant. These relationships have been shown elsewhere in TBI and dementia [10,12,13]. Fatigue was not related to CR. 

The logistic regression tested the hypothesis that CR (high versus low TOPF scores) would be strongly associated with neurocognitive scores post-SCI, even after adjusting for age, mental health, and fatigue. The data suggest that CR and neurocognitive outcomes are strongly and positively associated with an odds ratio of 5.4, even after adjusting for other factors that may influence the relationship. Age and depressive mood were both found to have a negative relationship after adjusting for the other factors. That is, as TOPF scores decrease, the likelihood is that depressive mood and age will increase. The finding of depressive mood is important given that evidence strongly suggests that depression and other psychosocial barriers (e.g., poor self-efficacy) can be substantial challenges to adjustment after an SCI [30,31]. However, it was surprising that TOPF scores were positively associated with anxiety (GADS-7) with an odds ratio of 1.69, and this seems to contradict previous correlation analyses. It was hypothesised anxiety would have a negative relationship with CR. That is, a low TOPF score is more likely to be related to a higher anxiety score (as found in the crude correlational analysis). Further research will be needed to clarify this finding.

### Study Limitation

The main limitation of this study is the lack of sufficient numbers in the longitudinal assessments at discharge and 12 months post-SCI (due to the COVID-19 pandemic). The study is still recruiting and continuing follow-up assessments to address this limitation.

In light of the COVID-19 restrictions, we accommodated the circumstances by conducting the NUCOG assessment via teleconferencing when necessary. However, it is essential to recognize that the adjustments made to test administration methods have the potential to influence the outcomes of the test. Specifically, participants’ level of attention during teleconferencing-based testing may differ from that observed in in-person assessments. Consequently, it is crucial to consider the potential implications of these adjustments on the reliability and validity of the test results, particularly in relation to participants’ attention levels. To address this limitation, we intend to conduct a comprehensive analysis in our future studies to assess any differences that may arise between in-person and remote administration methods.

Our study did not specifically focus on the direct effects of the pandemic. While we acknowledge that the pandemic may have influenced our results, it is challenging to disentangle its specific impact from the broader factors that might influence mental health post-SCI. Future research should explicitly focus on studying the direct effects of the pandemic on mental health and cognitive functioning to provide a more comprehensive understanding of these complex relationships.

## 5. Conclusions

These findings do provide helpful preliminary directions for improving the diagnosis of MCI after SCI. The relationship between CR and neurocognitive outcomes, after adjusting for potentially influential factors, has not been investigated previously after an SCI. The authors believe that the findings will lead to advances in SCI rehabilitation, such as the implementation of improved processes for assessing MCI after SCI, especially within the first 6 months post-injury. The findings also suggest that relying on a neurocognitive screen test conducted at one point in time is not sufficient, given the results of the logistic regression, where multiple factors were found to have a role in the relationship between CR and post-SCI cognitive outcomes. Lastly, the findings do indicate that CR/pmIQ should become a standard component of assessing neurocognitive status in adults with an SCI. These measures can help identify adults who may be at risk for cognitive decline and can be used clinically to evaluate the effectiveness of interventions aimed at enhancing CR.

## Figures and Tables

**Table 1 jcm-12-04258-t001:** Parametric and breakdown statistics for socio-demographic (*n* = 75), injury characteristics, TOPF, NUCOG, PHQ-9, GADS-7, and FSS for 65 participants.

Variables	All Participants	Probable MCI (NUCOG ≤ 88)	No MCI (NUCOG > 88)
Sex (males, *n* (%))	58 (77.3)	16 (84.2)	42 (75)
Age in years, mean (SD)	51.0 (18.3)	50.3 (21.9)	51.4 (17.5)
Years of education, mean (SD)	14.5 (3.1)	12.5 (3.2)	15.1 (2.9) *
Paraplegia, *n* (%) ^1^	34 (45.3)	--	--
AIS ^1,2^, *n* (%)			
*Grade A*	15 (20.5)	--	--
*Grade B*	7 (9.6)	--	--
*Grade C*	15 (20.5)	--	--
*Grade D*	35 (48.0)	--	--
*Grade E*	1 (1.4)	--	--
TOPF, mean (SD)	104.5 (14.3)	95.3 (13.4)	107.2 (13.7) **
NUCOG, mean (SD)			
*At admission*	91.2 (6.9)	81.5 (6.5)	94.5 (2.7)
*At discharge* ^3^	92.5 (4.9)	85.2 (3.6)	94.6 (2.8)
*At 12 months* ^4^	93.7 (5.3)	82.0 (5.3)	95.1 (3.3)
PHQ-9 at admission, mean (SD)	5.8 (5.0)	8.0 (7.0)	4.6 (3.7) *
GADS-7 at admission, mean (SD)	4.5 (4.2)	6.6 (5.8)	3.5 (2.9) **
FSS at admission, mean (SD)	32.4 (13.5)	32.0 (15.7)	32.7 (12.2)

Abbreviations: AIS: American Spinal Injury Association Impairment Scale; FSS: Fatigue Severity Scale; GADS-7: General Anxiety Disorder-7); NUCOG: The Neuropsychiatry Unit Cognitive Assessment Tool; PHQ-9: Patient Health Questionnaire-9; SD: Standard deviation; TOPF: Test of Premorbid Functioning. * *p* < 0.01 ** *p* < 0.01. ^1^ Two missing values for level of injury and AIS Grades: *n* = 73. ^2^ AIS grade definitions: Grade A: Complete: no motor or sensory function left below the lesion; Grade B: Incomplete. Sensory function, but not motor function, is preserved below the lesion, and some sensation is preserved in the sacral segments S4 and S5; Grade C: Incomplete. Motor function is preserved below the neurologic level, but more than half of the key muscles below the neurologic level have a muscle grade less than 3 (i.e., they are not strong enough to move against gravity); Grade D: The impairment is incomplete. Motor function is preserved below the neurologic level, and at least half of the key muscles below the neurologic level have a muscle grade of 3 or more (i.e., the joints can be moved against gravity); Grade E: The patient's functions are normal. All motor and sensory functions are unhindered. ^3^
*n* = 41; ^4^
*n* = 29.

**Table 2 jcm-12-04258-t002:** Breakdown of classification and misclassification of MCI versus no cognitive impairment for admission to rehabilitation (one missing value).

TOPF	MCI (*n*), ≤88	No MCI (*n*), >88	Total
≤101	10	17	27
>101	4	33	37
Total	14	50	64

Abbreviations: MCI: mild cognitive impairment; TOPF: Test of Premorbid Functioning. Χ^2^ = 6.28, df = 1, *p* < 0.05; odds ratio: 4.85; 95% confidence interval (CI): 1.32–17.79; *p* < 0.01. Sensitivity: 71.4%; 95% CI: 41.9–91.6%; specificity: 66.0%; 95% CI: 51.2–78.8%.

**Table 3 jcm-12-04258-t003:** Showing Pearson correlations between the study variables and TOPF.

Variable	TOPF Standardised Score
Age	0.52 **
Sex	0.14
Years of education	0.39 **
Level of injury	−0.15
NUCOG	
*At admission*	0.34 **
*At discharge*	0.43 *
*At 12 months*	0.54 **
PHQ-9, at admission	−0.16
GADS-7, at admission	−0.38 **
FSS, at admission	0.18

Abbreviations: FSS: Fatigue Severity Scale; GADS-7: General Anxiety Disorder-7); NUCOG: The Neuropsychiatry Unit Cognitive Assessment Tool; PHQ-9: Patient Health Questionnaire-9; TOPF: Test of Premorbid Functioning. * *p* < 0.05, ** *p* < 0.01.

**Table 4 jcm-12-04258-t004:** Logistic regression results for the association between the dependent binary TOPF variable (≤101 versus >101) with independent variables binary NUCOG (≤88 versus >88), age, mood (PHQ-9), anxiety (GADS-7), and fatigue (FSS).

Variable	Odds Ratio	95% CI	*p*	Log LRT	Χ^2^	*p*
NUCOG MCI	5.4	1–29	0.05	−30.5	4.2	0.04
Age	0.95	0.92–0.99	0.02	−38.8	8.4	0.004
PHQ-9	0.76	0.56–1.0	0.06	−32.9	4.3	0.04
GADS-7	1.69	1.1–2.5	0.02	−35.1	7.5	0.01
FSS	0.98	0.93–1.0	0.57	−32.6	0.69	0.40

Abbreviations: CI: confidence interval; FSS: Fatigue Severity Scale; GADS-7: General Anxiety Disorder-7); LRT: Likelihood ratio test; MCI: mild cognitive impairment; NUCOG: The Neuropsychiatry Unit Cognitive Assessment Tool; *p*: probability; PHQ-9: Patient Health Questionnaire-9; TOPF: Test of Premorbid Functioning; Χ^2^: chi-square.

## Data Availability

The datasets generated are available from the corresponding author on reasonable request.

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
