# Peer review of "Cognitive Reserve and Its Association with Cognitive and Mental Health Status following an Acute Spinal Cord Injury"

_jcm, 2023, doi:10.3390/jcm12134258_

Round 1

Reviewer 1 Report

Thank you for the opportunity to provide a peer review for this interesting article. This manuscript discusses an interesting area of research on cognitive reserve and mild cognitive impairment after acute SCI.

This study has a good comparison between psychological and cognitive measures. It would be more convincing if an MRI of the brain and SCI were provided to show the association with injury severity.

The language used in this study is generally good.

Reviewer 2 Report

Authors evidenced issues of cognitive decline after spinal cord injury that can have major impact in rehabilitation outcomes and should be thoroughly investigated in order to promote individualized healthcare strategies.  

The longitudinal analysis and correlation of CR/pmIQ and cognitive outcomes, with resource to TOPF and NUCOG, in individuals with SCI is novel and the study conlusions support the importance of assessing these factors routinely in SCI rehabilitiation.

However, there are some aspects of the present work that need further attention, such as:

- The age range of individuals in the study is big and this can induce variability in the mental health assessments. Are TOPF and NUCOG adapted according to age?

- Authors stated that due to Covid-19 restrictions NUCOG was administered via teleconferencing, when required. What is the influence of these adjustments in the test outcomes? For example, is it expected that participants will have the same level of attention while being tested in person vs via teleconferencing?

- The low sensitivity and specificity of the tests are pointed as issues in MCI misclassification post-SCI. How can this be overcome? What suggestions do authors have to counteract these findings? 

- Mood, anxiety and fatigue states are extremely variable and can significatly change even in a 24 hours period. Do authors consider that a single assessment at each of the timepoints analysed for each patient is enough to conclude about their overall mood, anxiety and fatigue levels? Shouldn't these factors be assessed in a longer period of time?

- Covid-19 was proved to have significant impact in the mental health of the population. How could the pandemic have influenced the results of this study? 

- The 12 months assessments will be crucial to validate these findigns. Authors should consider including preliminary data to support the conclusions.

Round 2

Reviewer 1 Report

Thank you for your response to the previous comments. What you mentioned in your response, including EEG, fNIRS, and brain Doppler techniques, are good to further support your conclusions. The reviewers believe it is worthwhile to include this evidence in the manuscript. Also, the second reviewer's comment about the COVID situation is worth focusing on, which may have a direct impact on the outcome of this study.

Reviewer 2 Report

The manuscript was improved in response to reviwers' comments and now contains important information about the influence of Covid-19 in the assessments and how this was taken into account. 
